# Face Image Encryption Based on Feature with Optimization Using Secure Crypto General Adversarial Neural Network and Optical Chaotic Map

**DOI:** 10.3390/s23031415

**Published:** 2023-01-27

**Authors:** Majed Alsafyani, Fahad Alhomayani, Hatim Alsuwat, Emad Alsuwat

**Affiliations:** 1Department of Computer Science, College of Computers and Information Technology, Taif University, Taif 26571, Saudi Arabia; 2Department of Computer Engineering, College of Computers and Information Technology, Taif University, Taif 26571, Saudi Arabia; 3Department of Computer Science, College of Computer and Information Systems, Umm Al-Qura University, Makkah 24382, Saudi Arabia

**Keywords:** information security, biometrics, face feature encryption, image optimization, cryptography, deep learning

## Abstract

Demand for data security is increasing as information technology advances. Encryption technology based on biometrics has advanced significantly to meet more convenient and secure needs. Because of the stability of face traits and the difficulty of counterfeiting, the iris method has become an essential research object in data security research. This study proposes a revolutionary face feature encryption technique that combines picture optimization with cryptography and deep learning (DL) architectures. To improve the security of the key, an optical chaotic map is employed to manage the initial standards of the 5D conservative chaotic method. A safe Crypto General Adversarial neural network and chaotic optical map are provided to finish the course of encrypting and decrypting facial images. The target field is used as a "hidden factor" in the machine learning (ML) method in the encryption method. An encrypted image is recovered to a unique image using a modernization network to achieve picture decryption. A region-of-interest (ROI) network is provided to extract involved items from encrypted images to make data mining easier in a privacy-protected setting. This study’s findings reveal that the recommended implementation provides significantly improved security without sacrificing image quality. Experimental results show that the proposed model outperforms the existing models in terms of PSNR of 92%, RMSE of 85%, SSIM of 68%, MAP of 52%, and encryption speed of 88%.

## 1. Introduction

DL is a very powerful technique in computer vision applications. Surveillance is a high-potential application field for DL. DL requires a large training data set to achieve excellent performance. However, collecting enough training data while maintaining the anonymity of people in data is expensive, especially for surveillance applications. No one wants their images to be included in the dataset because developers can monitor anyone’s actions. A comparable circumstance can be found during a surveillance operation. A security camera owner can monitor anything. Image encryption is one of the methods for maintaining privacy [1]. The picture encryption process converts an original image into an encrypted image in which no one can recognize the contents. Image encryption methods were primarily created to send photos securely over a public network. For people and machines to recognize the contents of an encrypted image, the image must first be decrypted. Anyone may recognize the contents of the image once it has been decrypted. This means that picture decryption can compromise privacy [2].

People have entered the era of big data as a result of technological advancements. Internet and computer technology have advanced swiftly, network popularity has grown significantly, and information interaction technology has matured. While most people utilize the Internet to send information, it also creates many data security risks. Data transmission security is rapidly affecting the security of individuals, businesses, and even countries as networks expand into new domains. Because of its visual features, an image has a strong expression influence on the data it contains. Many information expressions favor visuals because of their widespread use in information interaction. Owners of valuable photographs frequently utilize the Internet to conduct an auction or post their image data. That strategy removes geographical limits such as geography, and it is convenient and quick, but it also saves money [3]. However, throughout the network transmission process, insecure elements of picture data provide an opportunity for malicious attacks, and original image data may be attacked, resulting in data leakage or destruction. The goal of the picture encryption technique is to enhance the security of image data, minimize the risk of data leakage and destruction, and ensure the secure transmission of original data. In a few cases, image data are encrypted before transmission. For example, before medical images may be transmitted over the Internet, they must be encrypted to safeguard patient privacy. Criminal attacks such as destructive damage and data theft are common, and picture encryption technology is continually evolving. An urgent topic to be tackled is how to increase picture security, key transmission security, and anti-attack capability [4]. The iris, face, fingerprint, voice, deoxyribonucleic acid (DNA), and palm positions are examples of information properties created by human tissue structures. Because of their uniqueness, biological traits of the human body are extensively utilized to determine recognition and other sectors. The properties of the iris are extracted using iris recognition technology. It is part of the human biological feature extraction technology and is of extremely high grade. It is better for picture encryption because it improves the algorithm’s security and anti-attack capability. The identity recognition method based on iris feature extraction is receiving increased attention in academic and industry sectors. It has an extensive variety of applications and is progressively implemented in various departments with high security needs, such as finance and secrecy [5]. Applying DL techniques to the area of image security to resolve classic challenges has also received much attention recently and has made significant progress. However, many researchers are interested in how to better utilize the benefits of DL in image cryptography, image authentication, and image steganography. To assist relevant researchers in better understanding the field of DL uses in digital image security together with its upcoming progress, the origin and development method of DL techniques in image cryptography, steganography, and authentication were organized from numerous perspectives in this paper, as shown in Figure 1. We then evaluate these strategies, assess their benefits and drawbacks, and make recommendations for future research on this subject.

Various security techniques are currently available to assist in repelling picture-based attacks, but they are not effective in balancing security and image quality demands. Aside from that, chaotic map behavior provides a high level of security. As a result, combining DL and chaotic behavior can provide a superior picture encryption solution. As a result, the suggested article proposes a model in which a DL chaotic map is employed to perform better optimization to improve picture encryption performance. In computer vision applications, DL is a very powerful technology. Surveillance is a high-potential application field for DL. DL requires a large training data set to achieve excellent performance. However, collecting enough training data while maintaining the anonymity of people in the data is expensive, especially for surveillance applications. No one wants their images to be included in the collection because developers can monitor anyone’s actions in real-time. During a surveillance operation, a similar predicament can be encountered. A security camera owner can monitor anything. Image encryption is one of the methods for maintaining privacy. Image encryption converts an original image into an encrypted image in which the contents of the original image are unrecognizable. Image encryption methods were created primarily for the purpose of securely transmitting photos over a public network. For people and/or machines to recognize the contents of an encrypted image, the image must first be decrypted. Anyone may recognize the contents of the image once it has been decrypted. This means that image decryption has the potential to infringe on one’s privacy [6].

The contributions of this research are as follows:To propose a novel technique for face feature encryption with image optimization using cryptography and deep learning architectures;To develop a secure Crypto General Adversarial neural network and optical chaotic map for encryption and decryption of face images with optimization of images.

The rest of this research article is organized as follows. Section 2 of this paper shows the related work. In Section 3, novel techniques for face feature encryption with image optimization using cryptography and deep learning structural design are deliberated. In Section 4, experimental analysis and discussions are displayed. Section 5 of this paper contains its conclusion.

## 2. Related Work

This section contains traditional study projects that are only focused on image security. Furthermore, the literature uses picture encryption using chaotic methods [7]. However, achieving a suitable balance between security efficiency and encryption significance is difficult with these methods. DL, which uses multilayer neural networks (NNs) to extract features from raw input photos, has also attracted much interest in solving the problem. The advantages of convolution neural networks (CNNs) [8] are established in computer vision applications and picture domain transfer [9]. Image transfer from one domain to another is thought of as a texture transfer issue, to learn mapping connection amid an input image and output image from a set of matched image pairs. The most common image-to-image conversion approach is the cycle-reliable adversarial system [5], which offers two-cycle reliability losses that shift the image from one domain to another and then rebuilds back to the original image. DL technique is used to manage the image-denoising problem [10]. Image noise is interference data in image data that cause some useful image data to become invisible. The process of image denoising is considered image restoration [11]. The authors of [12] proposed a new cross-image, pixel scrambling-based rotation domain, dual-image encryption technique. In [13], the authors suggested a dual-image encryption method that incorporates DNA spatiotemporal chaos, deletion, and insertion to improve the security of the encryption process, and scramble real as well as imaginary sections of data produced by every round of encryption [14]. The approach encrypts two images simultaneously by combining DNA sequence insertion and deletion methods with scrambling and diffusion methods. Reference [15] introduced a new dual-image compression encryption method that improves the secrecy and resilience of the dual-image encryption approach. These techniques can employ the same encryption method to encrypt two photos separately, which requires decryption twice to extract the two images [16]. The authors of [17] proposed a method for medical picture cryptography based on a combination of chaotic and neural networks in their study. The major goal of the proposed method is to verify the safety of medical photographs using a less sophisticated method than current methods. Test findings supported the proposed method’s performance and efficiency, which meets digital imaging and communications in medicine criteria. In [18], the authors proposed a new multikey compressed sensing and ML privacy-preserving computing system. A user, a cloud, and a trusted third party make up this computer architecture, and the trusted third party is in charge of distributing random compressed sensing keys. In [19], a machine learning technique was used for an issue involving health, commercial, or other sorts of sensitive data, which necessitates not only precise estimates. Because the cloud does not have access to keys required to decrypt the data, encryption assures that it remains private. Table 1 presents a summary of image encryption techniques, including their benefits and limitations.

Consider two scenarios: the training and operating phase. In both cases, the network will most likely require a simple image dataset (see [31] for complete details). Typically, simple images are used to train the network. The original plain images should be decrypted to train the network, even if the image collection is encrypted. The individual who trains the network is referred to as a trainer in this context. The data holder who holds the training dataset is frequently not the same as the trainer. The data holder cannot then provide the dataset to the trainer using those two existing approaches, because doing so would violate the data holder’s privacy policy. In the operational phase, the scenario is similar to that in the training phase. To detect or classify an object, the network requires a basic image. The images should be decrypted for the network even if the encrypted images are stored in the surveillance system. In this way, the operator who runs the networked surveillance system may always examine the original plain photographs. As a result, a new picture encryption challenge is presented here. The fundamental difference from the existing image encryption challenge is that the encrypted images have desired qualities. The algorithm should encrypt images against both humans and networks in the present image encryption challenge. In the image encryption challenge discussed here, encoded images should be encrypted for humans while the network can be trained on encoded images. This type of encryption is known as learnable image encryption. Learnable image encryption is capable of encrypting images for human use. It means that the data owner can give their dataset while being compliant with the privacy policy. Trainers can use encrypted photos to train directly. The development of networks is extremely beneficial because the data holder and the trainer can avoid privacy concerns. The learnable image encryption is also effective during the operation phase. Encrypted images are used to train the network. As a result, without decrypting original plain photos, the network can recognize or classify objects using directly encrypted images.

## 3. System Model

This section discusses a novel technique of face feature encryption with image optimization using cryptography and deep learning architectures. Here, the input face image has been processed and mapped using optical chaotic maps that are utilized for efficient encryption and decryption of the image. Then, the secure crypto general adversarial neural network was developed for encryption and the decryption method with image optimization. The overall proposed method is represented in Figure 2.

### 3.1. Digital Optical Chaotic Mapping (Op-Ch_M)-Based Digital Image Encryption Technique

Xiong et al. offer a new chaotic map-based digital picture encryption technique. The surname initials of the author, abbreviated as XZQ, are used for brevity. The XZQ algorithm’s encryption phases are listed below.

*Phase 1.* Select chaotic mapping to produce a chaotic sequence with beginning specifications of $x_{0}$, and a number sequence of $=\left(n_{1}{,\;n}_{2\;}{,\;n}_{3}{\ldots ,\;n}_{k}\right){,\;0\;<\;n}_{i}\;<\;n$ and 
$\sum_{i=1}^{k}n_{i}=n$ where n is the picture row size.

*Phase 2.* Create a double-precise chaotic sequence $\left\{x_{1},x_{2},\ldots ,x_{n}\right\}\;$ using a chaotic mapping $f\left(x\right)=\mu x\left(1-x\right)$ arranging n items in the real sequence set $\left\{x_{1},x_{2},\ldots ,x_{n}\right\}\;$ in increasing order to generate a systematic sequence $\left\{x_{1}^{{}^{\prime }},\bar{\left.x_{2}^{{}^{\prime }},\ldots ,x_{n}^{{}^{\prime }}\right\}}\right.$ to produce a permutation address set $\left\{t_{1},t_{2},\ldots ,t_{n}\right\};\;$ at this point, $t_{i}\;$ numbers in $\left\{1,2,\ldots ,n\right\}$; commute pixels in first row based on $\left\{t_{1},t_{2},\ldots ,t_{n}\right\}$, namely transposing pixels $t_{i},i=1,2,\ldots ,n$.

*Phase 3.* Set $x_{1}=x_{n+n}$, and redo the process in Phase 2 in residual rows; $n_{i}=n_{k}$, repeatedly utilize δ from $n_{1}\;$ reintroduced δ.

*Phase 4.* Perform the same modifications in the image’s rows to
$L_{G}=\min_{G}(E_{x∼\mathrm{pdata}(x)}\log(1-D(G(x))))$
to complete image encryption.

G network starts with a convolution stage to encode and compress pictures spatially, and useful characteristics retrieved in this phase are then utilized in the transformation that follows. Finally, a 7 × 7 convolution kernel exports the forecast. Furthermore, the decryption network F has a similar structure as encryption network G; encryption network G has successfully converted original patient images. Therefore, encrypted network G’s loss LG is given by Equation (1):(1)$$L_{\mathrm{reconstruction}\;}{=E}_{x∼p_{\mathrm{data}\;(x)}}{∥Y-X∥}_{1}$$

G stands for an encryption network, and D stands for a discriminator network. G loss aims to reduce discriminator network D’s success rate in detection ciphertext produced by encryption network G. Aside from encryption, another suggested technique is to make certain that the restored image retains the original image’s texture data even when it is encrypted. Reconstruction loss estimates dissimilarity between G(x) and the original image for every image x from domain X, x → G(x) → F(G(x)) ≈ x. L is calculated using Equation (2):(2)$$L_{\mathrm{reconstruction}\;}{=E}_{x∼p_{\mathrm{data}\;(x)}}{∥Y-X∥}_{1}{=E}_{x∼p_{\mathrm{data}\;(x)}}\sum_{i=1}^{n}\left|y_{i}{-x}_{i}\right|\;{=E}_{x∼p_{\mathrm{data}\;(x)}}\left(\left|y_{1}{-x}_{1}\right|+\ldots +\left|y_{i}{-x}_{i}\right|\right)$$

The primary value of the 5D conservative chaotic method is evaluated by using pseudo-random sequence LCon based on the technique given below:$$\left\{\begin{array}{c} x0=\frac{\sum_{l=1}^{x}\mathrm{Lcon}(l)}{\omega_{0}}{+\alpha }_{0}\\ y0=\overset{n}{\underset{l=1}{\sum }}\mathrm{Lcon}\left(l\right){\times \psi }_{0}{+\beta }_{0}\\ u0=\frac{\sum_{l=1}^{x}\mathrm{Lcon}\left(l\right)\times 0.48}{\varphi_{0}}{+y}_{0}\\ v0=\frac{\log2\sum_{l=1}^{x}\mathrm{Lcon}\left(l\right)\times 0.48}{\varphi_{0}}{+\delta }_{0} \end{array}\right.$$
where n∈N, and $n<d_{0}.\;I=1,2,\ldots ,n$ is the index value. Initial control specifications are: $\alpha_{0}=-0.16,\;\beta_{0}=5.52,\;\gamma_{0}=$ $-2.24,\;\delta_{0}=-1.2$. $\varepsilon_{0}=-0.3$. Initial scale coefficients are $\omega_{0}=40318,\;\psi_{0}=0.0004,\;\varsigma_{0}=-2176,\;\phi_{0}=-15140,\;\varphi_{0}=8.667$. By Equation (13), the initial values of the 5D conservative chaotic method are evaluated as $x_{0}=0.2536,\;y_{0}=7.0021,\;z_{0}=-2.0216,\;u_{0}=$ 2.5102, $v_{0}=-0.7109$.

Phase 2. Arbitrary sequences X, Y, Z, U, V are given by iterating the 5D conservative chaotic method, and arbitrary matrices $RM_{1},\;RM_{2},\;K_{1},\;K_{2},\;K_{3}$ are obtained by transformation:$$\left\{\begin{array}{l} {\mathrm{RM}}_{1}=\mathrm{mod}\left({X\times 10}^{15},\;256\right)\\ {\mathrm{RM}}_{2}=\mathrm{mod}\left({Y\times 10}^{15},\;256\right)\\ K_{1}=\mathrm{mod}\left({Z\times 10}^{15},\;256\right)\\ K_{2}=\mathrm{mod}\left({U\times 10}^{15},\;256\right)\\ K_{3}=\mathrm{mod}\left({V\times 10}^{15},\;256\right) \end{array}\right.$$

$RM_{1}\;$ and $RM_{2}\;$ are arbitrary stage masks utilized for optical encryption channels, and $K_{1},K_{2},K_{3}\;$ are the keys to the digital diffusion channel’s encryption.

Phase 3. Optical encryption is performed on the image with low-bit scrambling $PL_{P2}\;$ to obtain low-bit encrypted image $EL_{P2}$. The encryption technique is described by the equation below:$$\begin{array}{l} {\mathrm{EL}}_{P_{2}}(x,y)=\\ F_{P_{2}}{\mathrm{FP}}_{2}\left\{{\mathrm{FP}}_{n}F^{p}p_{y}\left[{\mathrm{PL}}_{p_{2}}{(x,y)\times \mathrm{RM}}_{1}\left(u,v\right)\right]{\times \mathrm{RM}}_{2}\left(u,v\right)\right\} \end{array}$$
where $F^{P_{x2}}F^{P_{y}}\;$ is fractional Fourier transform through order $p_{x}$ for the x-axis and order $p_{y}\;$ for the y-axis. The high-bit scrambled image with dynamic adaptive inverse diffusion PHP1 yields EHP1.
$$\left\{\begin{array}{l} {\mathrm{EH}}_{P1}(\tau )=\mathrm{bitxor}\left(K_{1}{(\tau ),\mathrm{PH}}_{P1}-\mu \right)\\ \mathrm{EH}(\tau +1)=\\ \;\mathrm{bitxor}\;\left(\;\mathrm{bitxor}\;\left({\mathrm{EHP}}_{P1}{(\tau ),\mathrm{PH}}_{P1}{(\tau +1),K}_{2}\left(\tau +1\right)\right)\right)\\ {\mathrm{EH}}_{P1}(\tau -1)=\\ \;\mathrm{bitxor}\;(\;\mathrm{bitxor}\;(\mathrm{EH}\\ \left.\left.P_{1}{(\tau ),\mathrm{PH}}_{P1}{(\tau -1),K}_{2}\left(\tau -1\right)\right)\right) \end{array}\right.$$
where μ is the dynamic diffusion control specification fixed by the user and τ is the dynamic diffusion direction control specification,
$$\tau =\mathrm{mod}\left(\frac{\overset{M}{\underset{1=1}{\sum }}\overset{N}{\underset{j=1}{\sum }}\left(\frac{K_{2}+255}{3}\right)}{3\times M\times N}\times 10^{16},M\times N\right),\tau \in (1,M\times N)$$


The choice to balance data in two ciphertext pictures can be made based on application needs. The option to delete the 4-bit information when the communication proportion is higher can also be considered. If the image’s details are sought, it is essential to stabilize the data of an image with a high 4-bit ciphertext and an image with a low 4-bit ciphertext to obtain C1 and C2. In this example, the image balance approach is illustrated.
$$C_{1}{,C}_{2}\leftarrow \left\{\begin{array}{c} {\xi 3=\mathrm{EH}}_{P1}\left(x_{1}{(i),y}_{1}\left(j\right)\right)\\ {\mathrm{EH}}_{P1}\left(x_{1}{(i),y}_{1}\left(j\right)\right){=\mathrm{EL}}_{p2}\left(z_{1}{(i),w}_{1}\left(j\right)\right)\\ {\mathrm{EL}}_{p2}\left(z_{1}{(i),w}_{1}\left(j\right)\right)=\xi 3 \end{array}\right.$$

Discriminator network D seeks to distinguish between translated samples by maximizing discriminator network D’s classification accuracy, which is the inverse of the encryption network G’s goal Equation (3):(3)$$L_{D}{=E}_{x∼\mathrm{pdata}(x)}{\mathrm{logD}(x)+E}_{x∼\mathrm{pdata}(x)}\log(1-D(G(x)))$$

The private key for encryption is the final specifications of network G. In contrast, the private key for decryption is the final parameters of network F. The following is the procedure for producing a privacy key: For encryption, every convolutional layer’s parameters are initially arbitrarily initialized as given: $W_{n}=\mathrm{random}\left[w_{n,1},w_{n,2},\ldots ,w_{n,j},\ldots \right]$ where $w_{n}$ is the nth convolutional layer. As a result, the encryption privacy key W is made up of all specifications of every convolutional layer, which is described as follows: W= consist $\left[W_{1},W_{2},\ldots ,W_{n},\ldots \right]$.

In addition to forward propagation, the BP method transfers network loss between convolutional layers. Improve performance by updating the parameters in each layer. The gradient descent is defined by Equation (4):
(4)$$\begin{array}{rl} \theta_{j} & =\theta_{j}-\alpha \vee J(\theta )\\ & =\theta_{j}-\alpha \frac{\delta }{\theta_{j}}J(\theta )\\ & =\theta_{j}-\alpha \frac{\delta }{\theta_{j}}\frac{1}{2m}\sum_{i=1}^{m}{\left(h_{\theta }\left(x^{i}\right)-y^{i}\right)}^{2}\\ & =\theta_{j}-\alpha \frac{1}{2m}\sum_{i=1}^{m}\frac{\delta }{\theta_{j}}{\left(h_{\theta }\left(x^{i}\right)-y^{i}\right)}^{2}\\ & =\theta_{j}-\alpha \frac{1}{2m}\sum_{i=1}^{m}2\frac{\delta }{\theta_{j}}\left(h_{\theta }\left(x^{i}\right)-y^{i}\right)\left(\frac{\delta }{\theta_{j}}\left(h_{\theta }\left(x^{i}\right)-y^{i}\right)\right)\\ & =\theta_{j}-\alpha \frac{1}{m}\sum_{i=1}^{m}\left(h_{\theta }\left(x^{i}\right)-y^{i}\right)\times \left(\sum_{i=1}^{n}\frac{\delta }{\theta_{i}}\theta_{i}x_{i}-\frac{\delta }{\theta_{i}}y^{i}\right) \end{array}$$


The procedure of creating a privacy key for decryption is the same as producing a privacy key for encryption, excluding that the decryption network’s initial input becomes the encryption network’s projected output. Furthermore, the reconstruction loss is the loss of the decryption network, as shown in Equation (5).
(5)$$L_{\mathrm{reconstruction}\;}=E_{x∼p_{\mathrm{data}\;}(x)}\overset{n}{\underset{i=1}{\sum }}\left|F\left(P\left(x_{i}\right)\right)-O\left(x_{i}\right)\right|$$


The encryption algorithm is as follows:


1.Calculate the H value by extracting the characteristic value of the image to be encrypted.

$$x_{i}=\mathrm{mod}\left(\left(\mathrm{abs}\left(x_{i}\right)-\mathrm{floor}\left(\mathrm{abs}\left(\left(x_{i}\right)\right)\right){\times 10}^{44},256\right)i=1,2,3,4\right.$$



2.To carry out the process, utilize initial chaotic value $x_{0}$ and H value, producing initial value x 0’ utilized in scrambling chaotic sequence $\left\{C_{i},i=0,1,\ldots ,M*N-1\right\}$ as explained in Figure 3.3.Arrange the chaotic sequence $C_{i}\;$ in descending order; the resulting sequence is $C_{i}^{{}^{\prime }}$. Calculate mapping matrix A for converting $C_{i}\;$ to $C_{i}^{{}^{\prime }}$, for example, $C_{i}^{{}^{\prime }}=A*C_{i}$. 4.To obtain the final encrypted image $G_{0}^{{}^{\prime }}$, utilize matrix A to scramble the image $G_{0}\;$ according to the pixel location. $G_{0}^{{}^{\prime }}.G_{0}^{{}^{\prime }}=A*G_{0}$. 5.Decryption method: Extract the characteristic value of the image to be decoded.

Step 1. Calculate the hyperchaotic system’s generated random sequence using Equations (6) and (7).
(6)$$x_{i}=\mathrm{mod}\left(\left(\mathrm{abs}\left(x_{i}\right)\;-\;\mathrm{floor}\left(\mathrm{abs}\left(\left(x_{i}\right)\right)\right){\times 10}^{44},\;256\right)i=1,\;2,\;3,\;4\right.$$
$$\mathrm{Obviously},\;x_{i}\in \left[0,\;255\right]$$
(7)$${\overset{↼}{x}}_{1}=\mathrm{mod}\left(\left(x_{1}{+x}_{2}{+x}_{3}{+x}_{4}\right),\;4\right)$$

Step 2. Encrypt the acquired row–column permutation matrix and select the appropriate combination from Table 2 based on $x_{1}\left[0.3\right]$.
(8)$$\begin{array}{c} C_{3\times (i-1)+1\;}{=P}^{\mathrm{rc}}{\;}_{3\times (i-1)+1}\;⊕{\;D}_{x_{1}}\\ C_{3\times (i-1)+2\;}{=P}^{\mathrm{rc}}{\;}_{3\times (i-1)+1}\;⊕{\;D}_{x_{2},}\\ C_{3\times (i-1)+3\;}{=P}^{\mathrm{rc}}{\;}_{3\times (i-1)+1}\;⊕{\;D}_{x_{\;3\;{}^{\prime }}} \end{array}$$
where $D_{x_{1},}D_{x_{2}}$, and $D_{x_{3}}$ are given in Equation (9):(9)$$\begin{array}{c} D_{x_{1}}=\mathrm{mod}\left(\left(B_{x_{1}}⊕{\;C}_{3\times (i-1)+1},\;256\right),\right.\\ D_{x_{2}}=\mathrm{mod}\left(\left(B_{x_{1}}⊕{\;C}_{3\times (i-1)+2},\;256\right),\right.\\ D_{x_{3}}=\mathrm{mod}\left(\left(B_{x_{1}}⊕{\;C}_{3\times (i-1)+3},\;256\right),\right. \end{array}$$

Obviously, $D_{x}\in \left[0,255\right]$, where t = 1, 2, ... relates i-th hyperchaotic iteration; signifies $\mathrm{XOR},P_{i},i=1,2,\ldots ,M\times N$ relates scrambled image’s pixel value; $B_{x_{1}},B_{x_{2}}$, and $B_{x_{1}}$ reflect the corresponding combinations in Table 2 selected based on $\bar{x_{1}},C_{i},i=1,2,\ldots ,M\times N$. Step 3. The encryption procedure is complete if all plaintexts have been encrypted; otherwise, proceed to Step 1. The encryption and decryption processes are comparable. First, build the same hyperchaotic sequence with the same parameters and beginning values, but replace it with Equation (10), as follows:(10)$$\begin{array}{c} P^{\mathrm{rc}}{\;}_{3\times (i-1)+1}{=C}_{3\times (i-1)+2}\;⊕{\;D}_{x_{2},}\\ P^{\mathrm{rc}}{\;}_{3\times (i-1)+1}{=C}_{3\times (i-1)+3}\;⊕{\;D}_{x_{3},} \end{array}$$

Then, according to $\left\{r_{i},i=0,1,\ldots ,M-1\right\}$ and $\left\{c_{j},j=0,1,\ldots ,N-1\right\}$, the matrix is inversely transformed, and the original image is restored as shown in Figure 4.

In contrast to the typical method of training, i.e., two methods as a generator and a discriminator, three NNs are used here. A pair of NNs act as generators, while a third acts as a modified discriminator. Three NNs will be:6.Encryptor: Plaintext and a shared key, both in binary sequence, are used to produce encrypted text.7.Decryptor: The encrypted text is used as input, and the shared key are used to produce an output of decrypted text.8.Eavesdropper: This only accepts the encrypted text as input, which means it intercepts text and decrypts it without the shared key.

The general adversarial network (GAN) has the following layers; the architecture of all three NNs is the same as the subsequent layers:Dense layer that is fully linked;Flatten layer;Convolutional layer.

We employed one dense, four convolutional, and one flattened layer. Strided convolution is utilized to replace the pooling layer. We employed strided convolutions instead of immediate downsampling. Activation functions used: binary sequences, 0 and 1, are used for encryption. To standardize the output of each layer in $\left[1,\;1\right]$, the tanh initiation is utilized, but the last layer is utilized for sigmoid activation.

The output of A on inputs FF and Key is represented by $A\left(\omega_{A},\;\mathrm{FF},\;\mathrm{Key}\right)$, the output of the Server is represented by $S\left(\omega_{\mathrm{Server}\;},\;\mathrm{EFF},\;\mathrm{Key}\right)$, and the output of B on input C is represented by $B\left(\omega_{B},\;\mathrm{EFF}\right)$.

The distance function d is also incorporated into the facial feature at the same time. This study uses $L_{2}$ distance
$d_{2}\left(\mathrm{FF},{\mathrm{FF}}^{{}^{\prime }}\right)=\sqrt{\sum_{i=1,N}{\left({\mathrm{FF}}_{1}^{n}-{\mathrm{FF}}_{2}^{n}\right)}^{2}}$
for a specific operation, where N is the length of the facial feature. The loss function for each instance of B is defined by Equation (11):(11)$$L_{B}\left(\omega_{A}{,\;\omega }_{B},\;\mathrm{FF},\;\mathrm{Key}\right)=d\left(\mathrm{FF},\;B\left(\omega_{B},\;A\left(\omega_{A},\;\mathrm{FF},\;\mathrm{Key}\;\right)\right)\right)$$

$L_{B}\left(\omega_{A},\omega_{B},\;\mathrm{FF},\;\mathrm{Key}\right)$ reflects the inaccuracy of B when facial features are FF and key are in Equation (12).
(12)$$L_{B}\left(\omega_{A}{,\omega }_{B}\right){=E}_{\mathrm{FF},\;\mathrm{Key}}\left(d\left(\mathrm{FF},\;B\left(\omega_{B},\;A\left(\omega_{A},\;\mathrm{FF},\;\mathrm{Key}\right)\right)\right)\right)$$

This research acquires the "best B" by reducing that loss, as shown in Equation (13):(13)$$O_{B}\left(\omega_{A}{,\;\omega }_{\mathrm{Server}\;}\right){=\mathrm{argmin}}_{\omega_{B}}\left(L_{B}\left(\omega_{A}{,\;\omega }_{B}\right)\right)$$

In a similar vein, this paper constructs a sample Server reconstruction error and applies it to a distribution of face characteristics and keys by Equation (14):(14)$$\begin{array}{c} L_{\mathrm{server}\;}\left(\omega_{A}{,\;\omega }_{\mathrm{Server}\;},\;\mathrm{FF},\;\mathrm{Key}\;\right)=d\left(\mathrm{FF},\;\mathrm{Server}\left(\omega_{\mathrm{Server}\;},\;A\left(\omega_{A},\;\mathrm{FF},\;\mathrm{Key}\;\right),\;\mathrm{Key}\;\right)\right)\\ L_{\mathrm{Server}\;}\left(\omega_{A}{,\;\omega }_{\mathrm{Server}\;}\right){=E}_{\mathrm{FF},\;\mathrm{Key}\;}\left(d\left(\mathrm{FF},\;B\left(\omega_{\mathrm{Server}\;},\;A\left(\omega_{A},\;\mathrm{FF},\;\mathrm{Key}\right)\right)\right)\right) \end{array}$$

The optimal values of $L_{\mathrm{server}\;}$ and $L_{B}$ are combined in this study to define the Server and loss function of A by Equation (15):(15)$$L_{\mathrm{AServer}}\left(\omega_{A}{,\;\omega }_{\mathrm{Server}\;}\right){=L}_{\mathrm{Server}\;}\left(\omega_{A}{,\;\omega }_{\mathrm{Server}\;}\right)\;{-\;L}_{B}\left(\omega_{A}{,\;O}_{B}\left(\omega_{A}\right)\right)$$

This combination illustrates the aim of A and Server to reduce Server rebuild faults while increasing "optimal B" rebuild errors. However, the following research discusses beneficial alternatives.

By reducing $L_{A\;\mathrm{Server}\;}\left(\omega_{A},\omega_{\mathrm{Server}\;}\right)$, this study obtains "Best A and Server" by Equation (16):(16)$$\left(O_{A}{,\;O}_{\mathrm{Server}\;}\right){=\mathrm{argmin}}_{\left(\omega_{A}\omega_{\mathrm{Server}\;}\right)}\left(L_{\mathrm{AServer}}\left(\omega_{A}{,\;\omega }_{\mathrm{Server}\;}\right)\right)$$

### 3.2. Secure Crypto General Adversarial Neural Network

In this subsection, we provide the following algorithm for securing the crypto general adversarial neural network.

   Require:   c, clipping parameter, $m_{t}$, batch size. $\beta_{1}\beta_{2}$, hyperparameters parameters; $n_{\mathrm{critic}\;}$ amount of generator iterations per critic iteration.   1. while θ has not joined do   2. for $t=0,\ldots ,n_{\mathrm{cricic}\;}$ do 3; for i=1, 2, 3,…,m do   3. Sample ${\left\{x^{\left(i\right)}\right\}}_{i=1}^{m}-P_{r}$.    4. Sample ${\left\{2^{\left(i\right)}\right\}}_{i=1}^{in}-p\left(z\right)$.   5. a random number $\varepsilon ∼U\left[0,1\right]$.   6. 
$\overset{↼}{x}\leftarrow G_{\theta }\left(z\right)$
   7. 
$\overset{`}{t}\leftarrow \varepsilon x+\left(1-c\right)\overset{↼}{x}$
   8. 
$L^{\left(i\right)}\leftarrow D_{w}\left(\vec{x}\right)-D_{w}\left(x\right)+\lambda {\left(\|\nabla_{\overset{`}{x}}D_{w}{\left(\overset{`}{P}\right)\|}_{2}-1\right)}^{2}$
   9. 
$g_{w}\leftarrow \nabla_{w}\left[\frac{1}{w}\overset{m}{\underset{i=1}{\sum }}f_{w}\left(x^{\left(i\right)}\right)-\frac{1}{m}\overset{w}{\underset{i=1}{\sum }}f_{w}\left(g_{e}\left(z^{\left(i\right)}\right)\right)\right]$
   10. 
$w\leftarrow w+\alpha \cdot \mathrm{RMSProp}\left(w,8w\right)$
   11. 
$w\leftarrow \mathrm{clip}\left(w,-c,c\right)$
   12. end for   13. end for   14. 
$w\leftarrow \mathrm{Adam}\left(\nabla_{w}\frac{1}{w}\overset{m}{\underset{i=1}{\sum }}L^{\left(i\right)},w,\alpha ,\beta_{1},\beta_{2}\right)$
   15. end for   16. Sample ${\left\{z^{\left(i\right)}\right\}}_{i=1}^{m}∼p\left(z\right)$.   17. 
$g_{\theta }\leftarrow -\nabla_{\theta }\frac{1}{m}\overset{m^{1}}{\underset{i=1}{\sum }}f_{w}\left(g_{\theta }\left(z^{\left(i\right)}\right)\right)$
   18. 
$\theta^{{}^{\prime }}\leftarrow \theta -\alpha \cdot \mathrm{RMSProp}\left(\theta ,g_{\theta }\right)$
   19. 
$\theta \leftarrow \mathrm{Adam}\left(\nabla_{\theta }\frac{1}{m}\overset{m}{\underset{i=1}{\sum }}-D_{\omega }\left(G_{\theta }(z)\right),\theta^{\prime };\alpha ,\beta_{1},\beta_{2}\right)$
   20. end while

A GAN consists of two neural networks: a generator network and a discriminator network. The generator network is trained to generate new samples that are similar to a target distribution, while the discriminator network is trained to distinguish between the generated samples and real samples from the target distribution.

In this algorithm, the generator network is represented by $G_{\theta }$, and the discriminator network is represented by $D_{w}$. The generator network is trained to generate samples, denoted by $\overset{¯}{x}$, from a noise distribution, $p\left(z\right)$, while the discriminator network is trained to classify samples as either real (from the target distribution $P_{r}$) or fake (generated by the generator network).

The training process alternates between updates to the generator network and the discriminator network. During each generator update, the generator network generates a batch of samples from the noise distribution, and the discriminator network is updated to classify these samples as fake. During each discriminator update, the discriminator network is updated to classify a batch of real samples and a batch of fake samples generated by the generator network. The algorithm also uses various hyperparameters, such as the clipping parameter c, the batch size m, and the Adam optimization parameters α, $\beta_{1},\;\mathrm{and}\;\beta_{2}$. These hyperparameters set and tuned through experimentation to achieve the best performance for the specific task and dataset. The root mean squared propagation (RMSProp) optimization algorithm is also used to update the weights of the networks. The training process continues until the generator network has converged. The threshold value in the algorithm is the "clipping parameter" (c). It appears in the line “$w\leftarrow clip\left(w,-c,c\right)”,$ which sets the values of w to be within the range $\left(-c,\;c\right)$. The value of c is specified as an input to the algorithm and determines the maximum magnitude for the values of w.

A GAN’s purpose is to predict the possible distribution of existing data and produce new data samples with the same distribution. Generator G’s ability to produce samples is improved by creating a minimax confrontation procedure between it and discriminator D. The GAN’s main purpose is to create a generator G from real-world data X. The model’s objective function is given by Equation (17):
(17)$$\underset{G}{\min}\underset{D}{\max}=E_{x∼P_{d}}\left[\left(\log(D(x)]+E_{z∼P_{z}}[(1-D(G(z)))]\right.\right.$$


The distribution and the appearance of an image become increasingly similar as those of the cover image when a discriminator is added to the steganography system. For example, the following is the loss function given by Equation (18):(18)$$L_{\mathrm{disc}\;}{=E}_{c∼P_{c}}{[\log\;D(c)]+E}_{c∼P_{c},s∼P_{s}}\left[\log(1-D(H(c,s))]\;\right.$$

Here, $P_{c}$ and $P_{s}$ are covers and secret image distributions, and $H\;\left(c,\;s\right)$ is the steg image produced by the generator. Throughout the training operation, the hiding network and the extracting network are optimized to minimize the original secret image s and the loss of image-extracted secret image s 0. To do this, we propose a novel cost function that is reduced to enhance the method, which is given by Equation (19):
(19)$$\mathrm{argmin}\frac{1}{n}\overset{n}{\underset{i=1}{\sum }}\left(1-\mathrm{SSIM}\left(c_{i},H_{\theta }\left(c_{i},s_{i}\right)\right)\right.\;H_{\theta }$$


### 3.3. Security Analysis

Both the encryption and decryption networks include 24 levels, with a total of 2,757,936 parameters for each network. A deeper resnet-50 design is used for the ROI mining network. The ROI-mining network’s structure is shown in Table 2. Chest X-rays [45] are the dataset. The proposed solution is implemented on Nvidia Giga Texel Shader eXtreme (GTX) 2080Ti graphics card. Each epoch of the model takes about 10 minutes to train the network.

#### 3.3.1. Analysis of Key Space

The difficulty of an exhaustive attack is computed by the size of the key space. The number of specifications for the DL network is a key space of the proposed encryption method in this study, with an overall of 2,757,936 specifications in experimentations. Every specification or key is a 32-bit floating point value amid 0 and 1, which are written as a decimal integer with 10 significant digits in the computer. As a result, the encryption model’s key space can be stated as (1010) 2757936. Attackers will find it difficult to break down the system, and it will be able to efficiently resist attacks.

#### 3.3.2. Key Randomness Analysis

With the same conditions, the encryption network is trained four times. As a result, the parameters of these four networks, namely Key A, B, C, and D, are used as encryption keys. These four photos are unmistakably distinct. The SSIM index between various photos is usually less than 0.1, indicating that there is very little similarity between them. According to the experiment, the privacy keys for the medical picture encryption network are completely distinct after each training since the neural network’s parameters are randomly initialized. As a result of these differences, separate encrypted images are produced, each of which is processed using a different encryption network. The premise is that DL network training is inherently unstable. In different training, different initialization parameters might lead to different end parameters. It is shown that the proposed technique is similar to OTP and that it may be classified as an OTP technique.

#### 3.3.3. Key Sensitivity Analysis

DL models, unlike typical encryption systems, spread errors among layers. A 3 × 3 convolution kernel is used to send the lth pixel in the Nth layer feature map to a nearby pixel in (N + 1)th layer during the convolution process. When a feature point is incorrect, it is transferred to the next layer’s 3 × 3 feature points. The inaccuracy of feature points will grow by two pixels for every layer as the depth of the convolutional network increases. This inaccuracy grows exponentially with the superposition of the deconvolution process in the upsampling process. The attacker is assumed to have the most privacy keys in this experiment. Only around 5% of crucial specifications are changed, which is considered an unknown part. The encrypted image is then sent to the network with new specifications, and the network is unable to convert the ciphertext image back to the original. This means that even if only 5% of specifications are modified, the privacy key will fail to properly encrypt or decode the medical image. In other words, breaking the proposed technique requires attackers to estimate at least 95% of correct key specifications in a key space containing (1010) 2757936.

#### 3.3.4. Histogram Analysis

To calculate the performance of the suggested encryption network, the original image and the encrypted image are given in Figure 5c. The pixel distribution in the original image and the encrypted image is considerably varied, according to the experiment. The original chest X-ray image’s pixel histogram has 57,600 * (240 * 240) pixels overall, with more than 30,000 pixels having a value of 0, and greater than 5000 pixels having a value of 255. The original image’s pixel dissemination is fairly intense. The distribution of encrypted medical photos, on the other hand, is more uniform, which aids statistical analysis.

#### 3.3.5. Entropy Analysis

In contrast to statistical attacks, encrypted image data entropy is considered a significant measurable dimension for methods to defend. Image data entropy is a statistical property of an image’s grayscale distribution. The encrypted image should resemble random noise in appearance, the grayscale distribution should be uniform, and the anticipated value should equal 8.

#### 3.3.6. Security Analysis under Various Adversary Models

Tests are being carried out to determine if an attacker can produce a key under three various adversary methods. Hidden Factors Leakage: Four potential network structures are investigated in this experiment: network A, network B, network C, and network D. The training environment remains unchanged. Utilizing trained network A, the original image is encrypted. To restore the original image, the ciphertext image is decrypted by a decryption network retrieved from network A, network B, network C, and network D.

#### 3.3.7. Network Architecture Leakage

Different hidden factors are used in this experiment to train encryption networks with the same network configuration. All training circumstances remain constant.

#### 3.3.8. Both Network Architecture Leakage and Hidden Factors

To produce networks A, B, C, and D, the network is trained four times in this experiment under the same hidden elements and training circumstances. Investigation compares the decryption performance of these four networks on identical ciphertext images to determine if specifications produced by each are unique. Using decryption keys B, C, and D, the image’s gray value distribution differs significantly from that of the image decrypted with the help of decryption key A. It is seen that one network’s encryption of a medical image prevents its decryption by utilizing specifications of another network under same training conditions. Even if method specifications are learned using the same network method and hidden factors, they will not be able to decode the image. Experiments reveal that even though both network design and hidden elements are leaked, and the network is trained under identical circumstances, the specifications of every network are completely dissimilar, i.e., secure keys.

## 4. Experimental Analysis

The experiments were carried out on a personal computer (PC) with an NVIDIA GeForce Tesla V100 32G graphics processing unit (GPU), Pytorch 1.1, and Python 3.7 as the experimental environment. The input dataset’s image size is 32 × 32 pixels. The proposed method block size is set to 4. After block adaptation networks [7,8], pyramidal residual networks were created. Plain pictures, integrated cat map image encryption [9], a naive blockwise pixel shuffle, and the suggested technique are all compared here.

ImageNet dataset was used to gather 80,000 training photos and 10,000 test images for training network methods in this study. The Adam optimization technique was utilized to automatically modify the learning rate in the training phase to optimize method specifications. Hyper specifications were adjusted to 0.65 and 0.85, with the initial learning proportion set at 0.0001. The maximum number of training iterations was set to 250, while number of images per batch was set at 64. The Labeled Faces in the Wild (LFW) dataset was used for training and testing. This is a regularly used facial recognition test set. Because the face photographs are all taken from real-life situations, recognition difficulty is heightened, particularly due to elements such as different positions, expressions, age, lighting, and occlusion. Photos of the same individual are rather different. Multiple faces may appear in certain photographs. Only the center coordinate face is chosen as the goal in these multiface images, with the others being background interference. The LFW dataset contains 13,233 face pictures. Each photograph contains the name of the individual depicted. There are 5749 persons, and most of them have just one photo. Each photograph is 250 × 250 pixels in size; most are color images, but some are black and white portraits.

The comparative analysis between the proposed and existing techniques is conducted on the two datasets, ImageNet dataset and LFW, in terms of PSNR, SSIM, RMSE, MAP, and encryption speed. These matrices are defined below.

Peak signal-to-noise ratio, or PSNR, measures how well the original data and the reconstructed data match in terms of quality. It is frequently used to assess the effectiveness of data compression and reconstruction methods in the field of image and video processing.

A measure that evaluates the similarity between two pictures is called SSIM (structural similarity index). It is predicated on the idea that structural information, such as the association between neighboring pixels, is easily perceptible by humans due to their highly developed visual system. Images are compared for structural similarities using SSIM.

The discrepancy between two sets of data is measured by RMSE (root mean squared error). It is employed to evaluate the discrepancy between values that a model predicts and values that are observed. Information retrieval and classification algorithm performance are assessed using the MAP (mean average precision) measure. In the field of computer vision, it is frequently used to assess how well object detection methods are working.

PSNR (peak signal-to-noise ratio) is usually expressed in decibels (dB). SSIM (structural similarity index) is a dimensionless value that is typically expressed as a decimal value between −1 and 1, where a value of 1 indicates that the two images are identical. RMSE (root mean squared error) is typically expressed in terms of the pixel values of the images. MAP (mean average precision) is a dimensionless value that is usually expressed as a decimal value between 0 and 1. Encryption speed is typically expressed in terms of the number of operations per second that can be performed by the encryption algorithm.

Table 2 presents a comparative evaluation between suggested and existing methods in face image encryption based on DL architectures. Here, the ImageNet and LFW face datasets are compared when the proposed Cry_GANN_OChaMap and existing CNN and IEA methods are applied. The parametric analysis was carried out regarding PSNR, RMSE, SSIM, MAP, and encryption speed. Parametric analysis for the ImageNet and LFW datasets are displayed in Figure 5 and Figure 6, respectively. Initially, for the ImageNet dataset, the proposed Cry_GANN_OChaMap obtained PSNR of 92%, RMSE of 85%, SSIM of 68%, MAP of 52%, and encryption speed of 88%, as shown in Figure 5a through (e) respectively; while the LFW dataset attained PSNR of 90%, RMSE of 89%, SSIM of 61%, MAP of 59% and encryption speed of 89%, as shown in Figure 6a through (e), respectively. From this analysis, it can be observed that the proposed technique attained optimal results in face encryption based on DL techniques.

The digital optic chaotic mapping technique is utilized as a preprocessing step for the proposed face feature encryption technique. Chaotic mapping is being used to process and map the input face image, which is then fed into the Crypto General Adversarial neural network for encryption and decryption. The use of chaotic mapping allows for efficient encryption and decryption of the image, and the combination of chaotic mapping with deep learning techniques allows for the development of a secure encryption and decryption method. The results demonstrate that the proposed technique for face encryption outperformed existing techniques.

## 5. Conclusions

This research proposed a novel technique in secure face image encryption based on DL architectures. Here, the input face image is processed and mapped using optical chaotic maps, which are utilized for efficient encryption and decryption of the image. Then, the secure Crypto General Adversarial neural network was developed for the encryption and decryption process with image optimization. The key is crucial to successful encryption and decoding in cryptography. The key’s security determines information security. The standard image encryption scheme is vulnerable to key sharing and repudiation attacks. If the key is excessively long, it is difficult to remember and simple to lose. As times demanded, biometric encryption technology arose to address the issue of key security. The key is produced using an individual’s biometrics, and then used with appropriate picture encryption methods to attain data encryption. Uniqueness, stability, nonaggression, and other criteria of biological features that can be encrypted should be met. Experimental analysis was carried out for various datasets; the proposed Cry_GANN_OChaMap technique resulted in PSNR of 92%, RMSE of 85%, SSIM of 68%, MAP of 52%, and encryption speed of 88%.

Future research will focus on how the system learns reliable encryption algorithms for asymmetric encryption. Additionally, it is thought that the encryption technique developed through adversarial training may be used for a larger range of data types, including audio and image data, in addition to character data security.

## Figures and Tables

**Figure 1 sensors-23-01415-f001:**
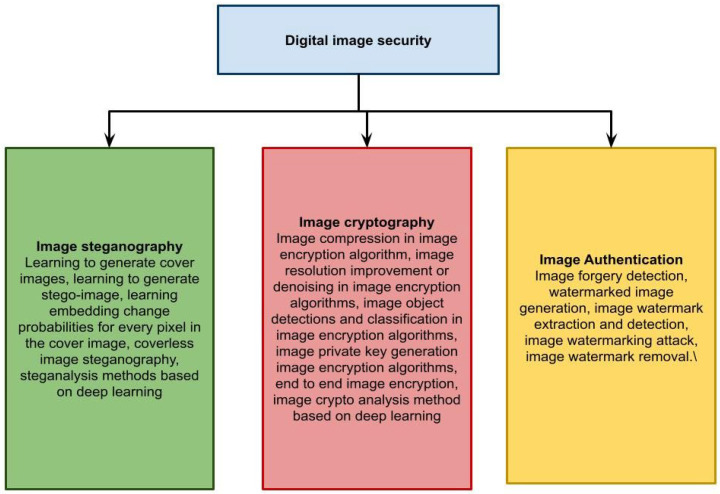
The entire architecture of the digital image security survey.

**Figure 2 sensors-23-01415-f002:**
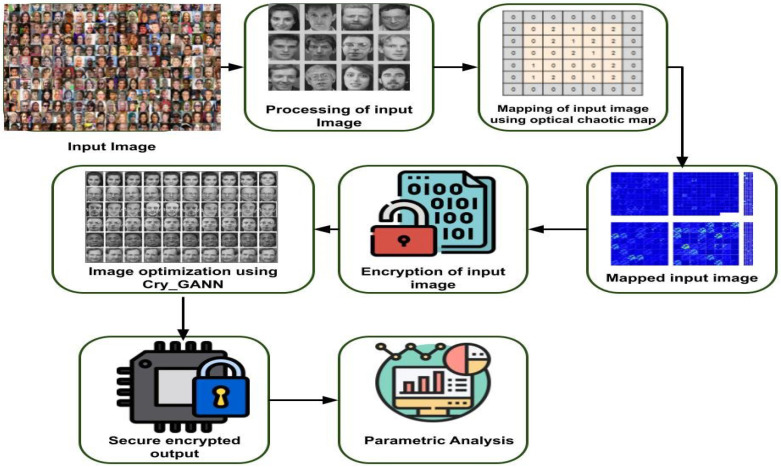
Overall proposed architecture.

**Figure 3 sensors-23-01415-f003:**
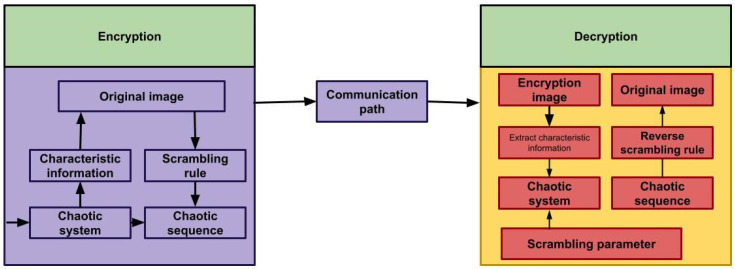
Schematic diagram for the chaos-based scrambling algorithm.

**Figure 4 sensors-23-01415-f004:**
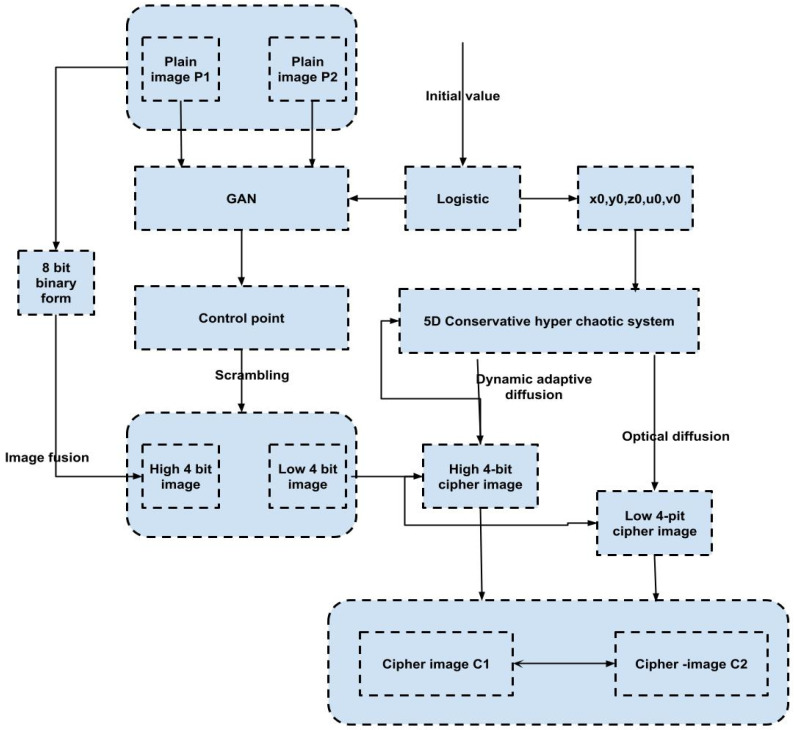
Flowchart of the encryption algorithm.

**Figure 5 sensors-23-01415-f005:**
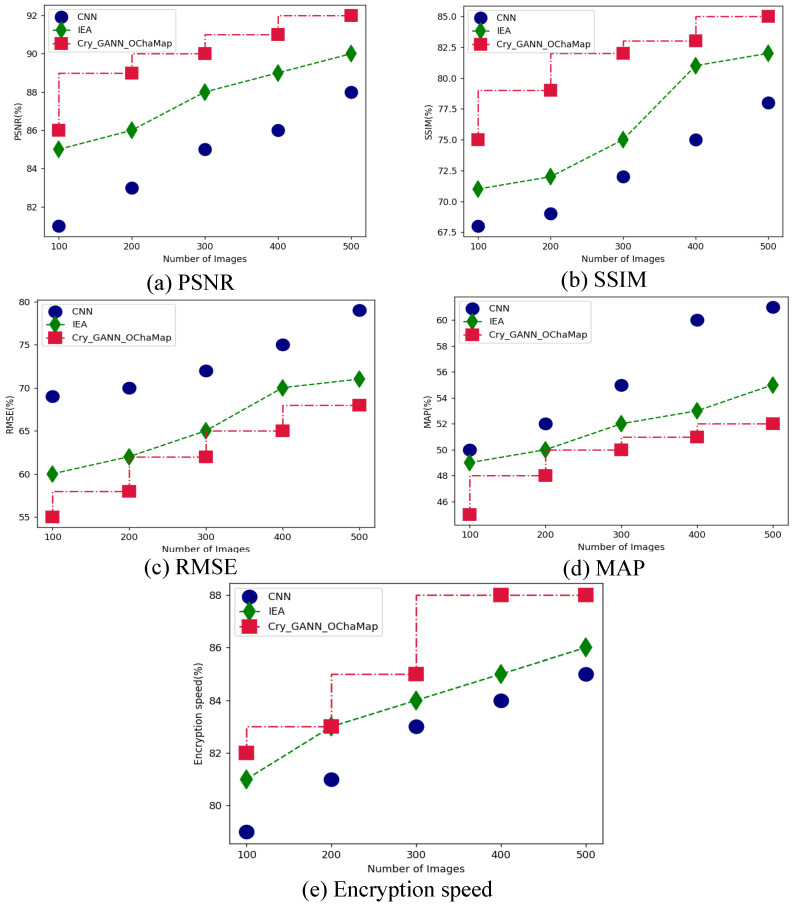
Comparative analysis between the proposed and existing methods for ImageNet dataset in terms of (**a**) PSNR, (**b**) SSIM, (**c**) RMSE, (**d**) MAP, and (**e**) Encryption speed.

**Figure 6 sensors-23-01415-f006:**
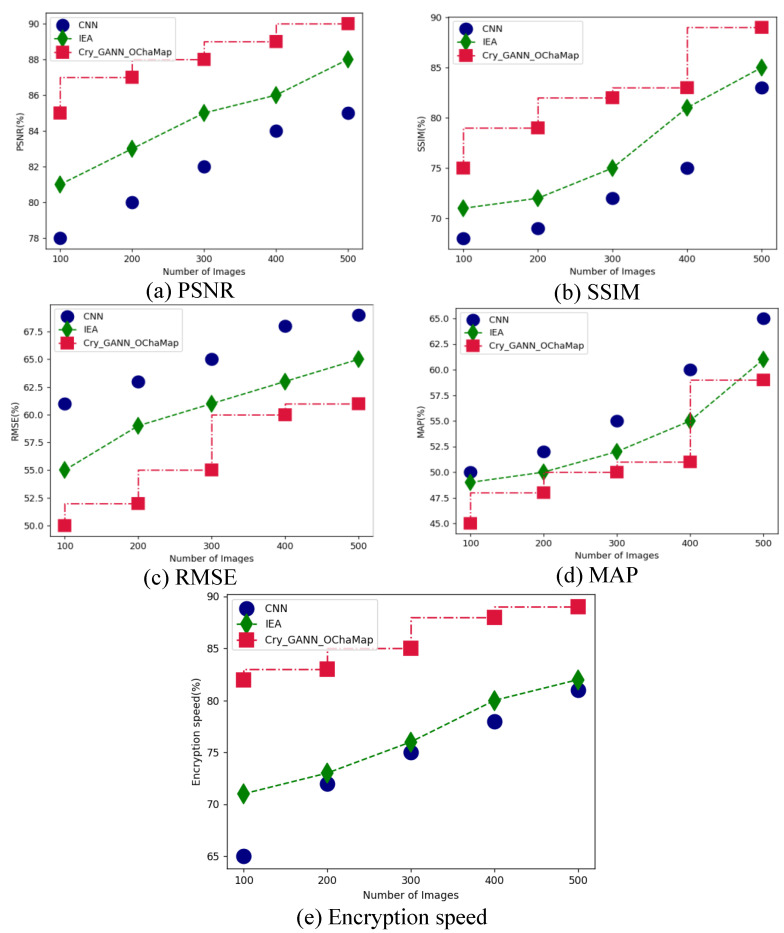
Comparative analysis between the proposed and existing techniques for the LFW dataset in terms of (**a**) PSNR, (**b**) SSIM, (**c**) RMSE, (**d**) MAP, and (**e**) Encryption speed.

**Table 1 sensors-23-01415-t001:** Different image encryption techniques.

Sr. No.	Image Encryption Technique	Overview	Advantages	Disadvantages
1.	Image encryption based on a public key [20]	Public key encryption uses a pair of keys, one for encryption and one for decryption, providing secure communication and nonrepudiation for image data.	Public key encryption provides secure communication as only the intended recipient can decrypt the image using their private key. It also allows for nonrepudiation and the ability to encrypt large amounts of data.	Public key encryption can be slower and more computationally expensive than symmetric key encryption. Additionally, managing and securely distributing the public and private keys can be complex and difficult.
2.	Chaos-based encryption technique [21,22,23]	Random starting circumstances. Numerous iterations are required; a sophisticated mapping process.	Chaos-based encryption techniques use chaotic systems to generate encryption keys, providing high levels of security and randomness. They also have the ability to resist known plaintext attacks and are resistant to differential cryptanalysis.	Chaos-based encryption techniques can be complex to implement and may have limitations in terms of encryption speed and scalability. They also may be sensitive to initial conditions and perturbations in the chaotic system.
3.	Visually meaningful image encryption technique [24,25]	Mentions an image that is at least twice as large as the original. A successful embedding method. A powerful encryption method.	Visually meaningful image encryption techniques help to preserve the visual features of an image while still encrypting it, making it more user-friendly and easy to understand. This also allows for more efficient and effective image transmission and storage.	Visually meaningful image encryption techniques may not provide as much security as other encryption methods and can be vulnerable to attacks such as stegonography and visual cryptanalysis. Additionally, it may be more computationally expensive and complex to implement.
4.	Partial image encryption techniques [26,27]	Extraction of important areas from images. Any safe encryption method.	Partial image encryption techniques allow for selective encryption of important or sensitive parts of an image, enhancing security while preserving the overall visual quality of the image. It also allows for more efficient storage and transmission as only certain parts of the image are encrypted.	Partial image encryption techniques may not provide as much security as full image encryption, as attackers may focus on the unencrypted parts of the image. It also may be more complex to implement and may require additional information to properly decrypt the image.
5.	Symmetric key encryption techniques [28,29,30]	Symmetric key confidentiality. Mechanism for safe key sharing and codec conformity.	Symmetric key encryption uses the same key for encryption and decryption, providing fast and efficient encryption. It also requires less computational power and is simpler to implement compared to other encryption methods, making it more practical for many use cases.	Symmetric key encryption requires secure key distribution and management, as the same key is used for encryption and decryption, if the key is compromised the security of the encrypted data is lost. It also does not provide nonrepudiation, meaning that the sender and receiver cannot prove who sent the message.
6.	Proposed encryption technique based on cryptography and deep learning (DL) architectures.	Here, the input face image is processed and mapped using optical chaotic maps, which are utilized for efficient encryption and decryption of the image.	The proposed encryption technique provides high security by combining the strengths of both methods. The technique can also adapt to changing encryption needs, improve the encryption efficiency, and resist attacks that traditional encryption techniques may fall prey to.	It may be computationally expensive and require specialized hardware and expertise to implement.

**Table 2 sensors-23-01415-t002:** Comparative analysis between the proposed and existing techniques.

Datasets	Techniques	PSNR	SSIM	RMSE	MAP	Encryption Speed
ImageNet dataset	CNN	88	78	79	61	85
	IEA	90	82	71	55	86
	Cry_GANN_OChaMap	92	85	68	52	88
LFW	CNN	85	83	69	65	81
	IEA	88	85	65	61	82
	Cry_GANN_OChaMap	90	89	61	59	89

## Data Availability

Not applicable.

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
