# Peer review of "Face Image Encryption Based on Feature with Optimization Using Secure Crypto General Adversarial Neural Network and Optical Chaotic Map"

_sensors, 2023, doi:10.3390/s23031415_

Round 1
Reviewer 1 Report
This study proposes a revolutionary face feature encryption technique that combines picture optimization with cryptography and DL architectures. To improve the security of the key, an optical chaotic map is employed to manage the initial standards of the 5D conservative chaotic method. A safe Crypto General Adversarial neural network and chaotic optical map are provided to finish the course of encrypting and decrypting facial images.
my comments are as follows:
1. The author-described technique has the ability to precisely segment the novelty. The Paper is written well and complied.
2. It would be nice if the author highlights more future directions of the proposed work.
3. The whole paper and References should be formatted accordingly.
4. The paper is clear and understandable but there are a lot of typos and grammatical mistakes. Always give proof-read to paper before submitting the final draft
5. some paper can be cited in introduction and background
Simple and Accurate Analysis of BER Performance for DCSK Chaotic Communication
Detection of image seam carving by using weber local descriptor and local binary patterns
Lightweight Single Image Super- Resolution Convolution Neural Network in Portable Device
Electrocardiogram soft computing using hybrid deep learning CNN-ELM
Author Response
Dear sir/madam
Thank you for giving us the opportunity to submit a revised draft of the manuscript “Face Image Encryption Based on Feature with Optimization Using Secure Crypto General Adversarial Neural Network and Optical Chaotic Map” for publication in the Journal of MDPI Sensor. We appreciate the time and effort that you and the reviewers dedicated to providing feedback on our manuscript and are grateful for the insightful comments on and valuable improvements to our paper. We have incorporated most of the suggestions made by the reviewers. Those changes are highlighted within the manuscript. Please see below, in blue, for a point-by-point response to the reviewers’ comments and concerns. All page numbers refer to the revised manuscript file with tracked changes.

Reviewer 2 Report
The paper seems to be very timely and focuses on a topic that is very paramount as it relates to encryption technique that combines picture optimization with cryptography and other deep learning architectures. The topic presented is good, but this paper still suffers from several issues as given below:
The manuscript is not well organized and written. Technical terms were not used. Authors should work on this aspect. There are too many short statements and they need to be rephrased. In line 1 the authors wrote “ML, often known as deep learning”. This statement is doesn’t seem correct. Authors should try and fix it. Similar issues in lines 38 and 183
2. The authors should improve on the literature review section of this manuscript. In section I, a table should be developed which compares the contributions of this work to other recent manuscripts on machine learning and DL based encryption techniques. The focus and coverage of the work, its limitations should also be included in the table.
3. This work lacks methodology. I cannot find the design framework. The methodology is missing, and this looks more like a review paper, where different concepts are introduced and discussed.
4. What is the novelty of this work? In section 3.1, the authors introduced “Digital Optic Chaotic Mapping” which has been developed by some other authors and available in the literature. Why did the authors bring this in? Are the results been used in the authors work? Please clarify the reason for this section 3.1. Also is section 3.1 linked with section 3.2.
5. The concept of PSNR, SSIM, RMSE, MAP used in Table 1 were not discussed and explained. Authors should explain each of this concept before use.
6. Results given in Figures 6a,6b, 6c, 6d, 6e are not properly explained. Please explain and discuss them
7. For the DL model architecture introduced in Section 3.2, the authors should provide the number of input layers, hidden layers, and the output layers. Also, some information on the hyperparameter tuning.
8. All the equations given especially in Section 3.2 should be properly explained. Also, the pseudocode and algorithms should be explained in a more concise but detailed manner.
9. More authoritative references should be cited.
Author Response

(The authors gave the same response as above.)

Reviewer 3 Report
Title : Face Image Encryption Based on Feature with Optimization Us- 2 ing Secure Crypto General Adversarial Neural Network and 3 Optical Chaotic Map ( may be cut short title’s number of words)
1. Careful editing required throughout your manuscript
For example 1: check this sentence.
This This section discusses novel techniques face feature encryption with image optimization using cryptography and deep learning architectures.
Example 2: what will be the expansion of ROI here
ROI network is provided to extract involved items from encrypted images to make data mining easier in a privacy-protected setting.
2. What are the reasons for taking 2 images at a time as per encryption process as illustrated in Figure 4.
3. What is the threshold value set if you are using GAN alongside encryption algorithm for images
4. List out the features taken for the encryption and decryption process involved for image security.
5. Provide mathematical expressions on various analysis for security. Just look 3.3 to its end.
6. It is better, if you are provided the parameters units for example PSNR SSIM RMSE MAP Encryption speed.
7. In conclusion, you may present future work to support your idea.
8. Citations are not sufficient to claim your article is good enough.
9. I don't know, what is the novelty of your proposed work? Can you explain after the results of your experiment!
Author Response

(The authors gave the same response as above.)

Round 2
Reviewer 2 Report
Some comments have been addressed by the authors while some have not been addressed.
The comments that have not been addressed are given below:
1. The authors should improve on the literature review section of this manuscript. In section I, a table should be developed which compares the contributions of this work to other recent manuscripts on machine learning and DL based encryption techniques. The focus and coverage of the work, its limitations should also be included in the table.
2. For the DL model architecture introduced in Section 3.2, the authors should provide the number of input layers, hidden layers, and the output layers. Also, some information on the hyperparameter tuning.
3. More authoritative references should be cited.
Author Response
Thank you for giving us the opportunity to submit a revised draft of the manuscript “Face Image Encryption Based on Feature with Optimization Using Secure Crypto General Adversarial Neural Network and Optical Chaotic Map” for publication in the Journal of MDPI Sensor. We appreciate the time and effort that you and the reviewers dedicated to providing feedback on our manuscript and are grateful for the insightful comments on and valuable improvements to our paper. We have incorporated most of the suggestions made by the reviewers. Those changes are highlighted within the manuscript. Please see below, in blue, for a point-by-point response to the reviewers’ comments and concerns. All page numbers refer to the revised manuscript file with tracked changes.

Reviewer 3 Report
Title : Face Image Encryption Based on Feature with Optimization Using Secure Crypto General Adversarial Neural Network and Optical Chaotic Map
Authors are revising this manuscript as per my comments. But however, I can see some more flaws. Those are as follows.
1. Deep Learning (DL) is repeated multiple times. Please use DL then and there after Deep Learning (DL).
2. In the abstract, you can mention the output parameters and their results as values. Since a good article can be understood with the abstract itself.
3. Please omit words like we, you etc. throughout this article.
4. You can expand the abbreviations such as GPU, PC, NVIDIA etc. once.
Author Response

(The authors gave the same response as above.)
